# Online Calibration Study of Non-Contact Current Sensors for Three-Phase Four-Wire Power Cables

**DOI:** 10.3390/s23052391

**Published:** 2023-02-21

**Authors:** Peiwu Yan, Wenbin Zhang, Le Yang, Wenying Zhang, Hao Yu, Rujin Huang, Junyu Zhu, Xi Liu

**Affiliations:** 1College of Mechanical and Electrical Engineering, Kunming University of Science and Technology, Kunming 650504, China; 2College of Electric Power, Kunming University of Science and Technology, Kunming 650504, China; 3College of Science, Kunming University of Science and Technology, Kunming 650504, China

**Keywords:** three-phase four-wire power cables, sensor arrays, reconstruction current, online self-calibration

## Abstract

Three-phase four-wire power cables are a primary kind of power transmission method in low-voltage distribution networks. This paper addresses the problem that calibration currents are not easily electrified during the transporting of three-phase four-wire power cable measurements, and proposes a method for obtaining the magnetic field strength distribution in the tangential direction around the cable, finally enabling online self-calibration. The simulation and experimental results show that this method can self-calibrate the sensor arrays and reconstruct the phase current waveforms in three-phase four-wire power cables without calibration currents, and this method is not affected by disturbances such as wire diameter, current amplitudes, and high-frequency harmonics. This study reduces the time and equipment costs required to calibrate the sensing module compared to related studies using calibration currents. This research offers the possibility of fusing sensing modules directly with running primary equipment, and the development of hand-held measurement devices.

## 1. Introduction

Smart grids have been developed for more than ten years now. Research and practice in smart grids at home and abroad have shown that advanced sensing technology is an essential element of smart grids [1,2,3]. South China Power Grid’s Li Lixin’s team of academicians put forward the concept of a transparent power grid. This transparent power grid will use sensor measurement technology as a supporting technology, and a wide range of sensors deployed in all aspects of the power system as its sensory layer and nerve endings; these are the physical basis and prerequisites for achieving the transparency of the power grid [4,5,6]. In low-voltage distribution networks, three-phase four-wire power cables, as one of the main ways of transmitting electrical energy in distribution networks of 1 kV and below, are widely used in scenarios such as factories, lighting, residential, and agricultural power consumption at the end of the distribution network [7]. Not only does this have the advantages of being a mature technology and having a stable transmission, but it is also an integral part of the grid sensing system that cannot be ignored. The low current voltage distribution network mainly uses mutual inductor technology, but the traditional current transformer is generally only used to measure the frequency current signal; high current measurement will lead to magnetic saturation, and it cannot measure the high-frequency signal, the device is bulky, and installation locations are limited [8]. It is obvious that conventional equipment is no longer sufficient to meet the critical and widely distributed needs of sampling nodes [9]. Information regarding the current flow is important for modern power systems and many industrial, commercial, and residential practices. Measuring the power conductor current accurately and easily is always a challenging task in the electric power industry [10,11].

Safe, accurate, and convenient testing of cable transmission lines, and the assessment of the operational status of power cables, is of great importance [12,13,14]. In recent years, therefore, non-contact sensor arrays have been used by many scholars for the measurement of currents in multi-core cables. The literature [7,15,16,17,18,19] presents studies using magnetic field measurements such as piezoelectric cantilever beam current sensors, Roche coils, or tunneling magnetoresistance combined with algorithms such as the Newton–Raphson method, the gradient method, or the trust domain to assist in solving the magnetic field decoupling matrix. All of the above studies use offline calibration currents as a prerequisite for calibration, so the studies are applied to out-of-service cables or unfinished facilities, i.e., they are not convenient for direct integration with primary equipment in service. The literature [20,21,22] applies pulse width modulation (PWM) to the load to create an identifiable carrier signal in the original current waveform. In contrast to offline calibration, the pulsed carrier can solve the magnetic field decoupling matrix without the cable being out of service. The strength of the carrier signal is related to the load parameters, and the cost of calibration increases with the amount of carrier in the cable to be measured. Application scenarios are given in this paper for current and power monitoring in ships, certain factory equipment, and domestic units.

To address the above problems, this paper proposes an online calibration method without calibration currents. The distribution of the tangential component of the magnetic field strength outside the cable is first simulated and analyzed, and then the distribution pattern of the magnetic field strength near the cores of different phases is generalized, based on which a new online calibration method and sensor array design are proposed. Compared to studies using calibration currents, this research reduces the time and equipment costs required to calibrate sensing modules for multi-core calibration. This also offers the possibility of integrating sensing modules directly with running primary equipment and handheld measurement devices.

## 2. Self-Calibration Theory and Simulation

### 2.1. Multi-Core Cable Measurement Principle

According to the magnetic field superposition theorem, the key to the current reconstruction of multi-core cables is to establish a one-to-one correspondence between the measured conductor and the sensor, that is, to determine the specific values of each element (gain coefficient) in the magnetic field decoupling matrix. The current waveform reconstruction equation of the three-phase, four-wire power cable is shown in Equation (1) [21].
(1)$$I_{3\times 1}=K_{3\times 3}^{-1}\cdot V_{3\times 1}$$
where $I_{3\times 1}$ indicates the three-phase current to be measured, $K_{3\times 3}$ is the complex magnetic field decoupling matrix, and since this equation is used to solve for transient magnetic fields, each gain coefficient in the matrix is a real number, and according to Kirchhoff’s current law, the most simple form of the matrix $K_{3\times 3}$ can be derived as a third order with nine gain coefficients. $V_{3\times 1}$ indicates the output of the magnetic sensor. As can be seen from matrix $V_{3\times 1}$, the minimum number of magnetic sensors used for the current waveform reconstruction in three-phase four-wire power cables, is three. The three magnetic sensors form a circular array, which not only simplifies the calculation but also facilitates the measurement of the magnetic field in the tangential direction. The array configuration adds a new dimension to the measurement and increases the opportunity to estimate unknown parameters. Figure 1 shows a schematic diagram of a three-phase four-wire power cable in relation to the location of the circular sensor array.

With the random distribution of cable and sensor positions in Figure 1, the response of sensor 1 to phase A current at this point can be derived as Equation (2)
(2)$${\dot{V}}_{A1}=\frac{\mu G\cos\theta }{2\pi \sqrt{{\left(x_{1}-x_{2}\right)}^{2}+{\left(y_{1}-y_{2}\right)}^{2}}}{\dot{I}}_{A}=k_{11}{\dot{I}}_{A}$$
where μ is the induced permeability (approximate air permeability), G is the sensor gain, and θ is the angle between the direction of magnetic field strength and the sensitive axis of the sensor. $\left(x_{1},y_{1}\right)$ is the coordinate of the center of the A-phase line, $\left(x_{2},y_{2}\right)$ is the coordinate of the center position of sensor 1, ${\dot{I}}_{A}$ is the phase A line current, and $K_{11}$ is the first gain coefficient in the decoupling matrix of the composite magnetic field. As can be seen from Equation (2), the key factors affecting the magnitude of the value of the gain coefficient in the matrix $k_{3\times 3}$ are the relative position relationship and the measured gain of the sensors. The location of the sensor array is known, so determining the location of each phase core in the cable without damaging the insulation is the key to the problem.

### 2.2. Tangential Component of Magnetic Field Strength Outside Cable

Figure 2 shows a diagram of the component of the magnetic field strength in the tangential direction (direction of the sensor sensitivity axis) at a given moment. “r” in Figure 2 is the distance from the center of the four-phase core to the center of the cable. The A, B, C, and N phase position coordinates are defined by “r”, respectively A (−r, 0), B (0, −r), C (0, r) and N (r, 0); a is the angle between the sensor center and the cable center line and the positive direction of the x-axis; R is the radius of the circular sensor array, the sensor position coordinates are (Rcosa, Rsina). Another n = R/r, I_A_ = I_B_ = I_C_ = I, according to the Ampere loop theorem the sensor sensitive axis direction of the magnetic field strength component amplitude H′ can be known.

It can be seen from Equation (3) (H′ (n, a)) that the amplitude of the magnetic field strength component in the direction of the sensor sensitivity axis is only related to the radius of the sensor array and the measurement angle a, if the size of the cable to be measured is determined; at the same time, due to spatial symmetry, the value of H is symmetrically distributed in the interval a (0°,180°) and the interval a (180°,360°), so the half-perimeter simulation can also be generalized to a one-week distribution law.
(3)$$H^{\prime }=\frac{I}{2\pi r}\sqrt{\begin{array}{l} {\left[\begin{array}{l} \left(\frac{n\sin a}{n^{2}+2n\cos a+1}-\frac{n\sin a+1}{2n^{2}+4n\sin a+2}-\frac{\left|1-n\sin a\right|}{2n^{2}+4n\sin a+2}\right)\sin a\\ +\left(\frac{\sqrt{3}n\sin a+\sqrt{3}}{2n^{2}+4n\sin a+2}-\frac{\left|\sqrt{3}-\sqrt{3}n\sin a\right|}{2n^{2}-4n\sin a+2}\right)\cos a \end{array}\right]}^{2}+\\ {\left[\begin{array}{l} \left(\frac{\left|n\cos a+1\right|}{n^{2}+2n\cos a+1}-\frac{\left|n\cos a\right|}{2n^{2}+4n\sin a+2}-\frac{\left|n\cos a\right|}{2n^{2}-4n\sin a+2}\right)\sin a\\ +\left(\frac{\left|\sqrt{3}n\cos a\right|}{2n^{2}+4n\sin a+2}-\frac{\left|\sqrt{3}n\cos a\right|}{2n^{2}-4n\sin a+2}\right)\cos a \end{array}\right]}^{2} \end{array}}$$

In order to understand the change law of magnetic field strength at *a* ∈ [0°, 180°], the simulation analysis is carried out for *H*′. The current used in the simulation is a sinusoidal current, with an RMS value of 5 A and a current frequency of 50 Hz, current frequency ω = 50 Hz, and the actual measurement of 4 × 6 mm^2^ cable (GB/T5013.4-2008), from which it can be found r ≅ 2.687 mm. Part of the simulation results are shown in Figure 3. Figure 3 shows the simulation results of the magnetic field strength on the interval of *a*∈[0°, 180°] when n = 3, 4.1, 5.6, 6.33, and 7.15 (these values are taken randomly).

Figure 3 shows that, although the magnetic field strength is non-linearly distributed, the curves have a finite number of extreme points and the monotonic intervals of the five curves show a similar distribution pattern. Table 1 shows the locations of the extreme “a” values of the curve shown in Figure 3; the extremes are noted counterclockwise as first, second, third and fourth in order.

The following points can be summarized after extensive simulation rules, as shown in Table 2.

To further determine the relationship between the variable “n” and the distribution of extreme values, Figure 4 shows the simulation results of the distribution of extreme values and monotone intervals as “n” varies.

The simulation chose the interval n ∈ [1.5, 10], the size of n reflects the distance between the sensor array and the wire core to be measured. Table 3 shows the distribution law of the eight extreme point locations.

The four-point law summarized in Table 2 is verified in the simulation of the extreme points and monotone interval distribution.

## 3. Calibration Scheme and Sensor Array Design

### 3.1. Calibration Scheme Design

As can be seen from the theoretical analysis in the previous section, the magnetic field strength of the cable is non-linearly distributed over a week, so it is not possible to directly determine the random position relationship between the sensor array and the cable, the exact value cannot be inferred, but the sensor array can be rotated to a specific position by means of eigenvalues, and from this position information the corresponding magnetic field decoupling matrix can also be established. The simulation results show that the magnetic field strength is minimal at a = 0°, and equal at a = 90° and a = 270°, regardless of the variation of n. The calibration process, therefore, revolves around these two eigenvalues.

The online calibration of the sensor array is divided into two steps: in step one, one sensor rotates for one week to find the minimum value of the magnetic field strength, aligning near phase N; in step two, two sensors, differing by 180°, find positions with equal values of the magnetic field strength in a small area, aligning to phases B and C, respectively. Since there are multiple monotonic intervals of magnetic field strength in the interval *a* ∈ [0°,180°], there is no guarantee that there will be only one pair of positions with equal values, so the order of the calibration steps must not be reversed.

As only specific frequency components are extracted for analysis, the higher harmonics in the cable do not interfere with the identification of the eigenvalues in the magnetic field distribution. They do not affect the online calibration process that follows. According to Equation (3), H′ increases linearly with the current amplitude, so the current amplitude to be measured only affects the magnetic field strength, which not only does not affect the magnetic field distribution characteristics of the cable for one week but also makes the eigenvalues easier to capture and the calibration process is less affected by the disturbing magnetic field, so the higher the current to be measured in the same cable, the more favorable the use of this online calibration method. Similarly, the geometrical parameters of the wire diameter do not affect the magnetic field distribution pattern.

### 3.2. Sensor Array Design

The value of n indirectly reflects the distance of the sensor from the core to be measured. When designing the sensor array, a small value of n will not only restrict the layout of the sensors in the array but also make the impact of dimensional accuracy on the measurement more obvious; a large value of n will not only make it more difficult to distinguish magnetic field strength variations but will also be detrimental to the alignment of the sensor array.

Combined with the calibration scheme, this paper gives the sensor array design scheme: ➀ at least three sensors; ➁ at least one pair of sensors separated by 180°; ➂ according to experimental experience, n ∈ [3, 8] is appropriate. The experimental sensor array design in this paper is shown in Figure 5, where sensor two is used for calibration step one, and sensors 1 and 3 are used for calibration step two.

The experimental current in this paper is low, so the fluxgate sensor array is used to measure the current in each phase, but this is not the only technical solution. This calibration method can be applied to Roche coils, reluctance, Hall, and other sensor arrays designed by different technical means to perform the measurement. Figure 6 shows the physical diagram of the sensor array in this paper.

The fluxgate sensor array is drawn as a two-layer PCB with three sensors sensitive axis to the center of the array 17.5 mm, chip DRV425RTJT, and calculated sensor measurement gain G = 4.88 V/mT. According to Equation (2), in Figure 5 the position relationship can be derived from the matrix K, as shown in Equation (4), unit mV/A.
(4)$$K=\frac{\mu G}{2\pi }\left[\begin{array}{ccc} \frac{R}{R^{2}+r^{2}} & \frac{1}{R+r} & \frac{1}{R-r}\\ \frac{1}{R+r} & \frac{R}{R^{2}+r^{2}} & \frac{R}{R^{2}+r^{2}}\\ \frac{R}{R^{2}+r^{2}} & \frac{1}{R-r} & \frac{1}{R+r} \end{array}\right]=\left[\begin{array}{ccc} 54.5 & 48.3 & 65.9\\ 48.3 & 54.5 & 54.5\\ 54.5 & 65.9 & 48.3 \end{array}\right]$$

### 3.3. Auxiliary Centering Device

After discarding the calibration currents, the acquisition of the magnetic field decoupling matrix relies on several geometrical parameters in the measurement system, which coaxially has a significant impact on the calibration, for which an auxiliary alignment device is designed to assist in subsequent experiments.

The auxiliary centering device is composed of the upper dial, lower dial, centering nut, support frame, and some standard thread parts. Figure 7 shows the main parts of the auxiliary centering device and its assembly relationship. The upper and lower dial and cable interference fit together, fixed by the standard thread parts, to ensure that there is no relative sliding between the dial and cable; the axis position of the upper and lower dials has half a conical bulge, which forms a complete conical surface after interference fit with the cable. The three fulcrums of the middle nut press the inner hole of the PCB board on the conical surface, so that the center of the sensor array is collinear with the center of the cable. The support bracket holds up the cable to ensure that the cable near the measuring point is straight. During the experiment, the pointer on the PCB board aligns the scale on the dial, then tightens the center nut, fixes the PCB board, collects the waveform, loosens the center nut, adjusts the position of the pointer, and tightens it again, so the experiment is repeated.

## 4. Experimental Verification and Analysis

### 4.1. Magnetic Field Strength Distribution Experiment in the Tangential Direction

To verify the reliability of the online calibration method, the following magnetic field distribution experiment was designed to verify that the magnetic field strength amplitude is minimized at a = 0° and that a = 90° and a = 270° are within the monotonic interval. The experimental procedure is centered on the cable, rotating the magnetic sensor array, saving three cycles (at least one cycle) of the waveform at each dwell position of sensor two, digitally filtering the waveform, and extracting the amplitude-frequency response 50 Hz eigenvalues, the specific experimental system is shown in Figure 8 below.

The magnetic field distribution experimental system consists of the three-phase current source, oscilloscope, three-phase four-wire power cable, three-phase resistor, sensor array, and auxiliary centering device. An auxiliary centering device is used to ensure that there is a high coaxial between the cable and the sensor array. In the experiment, the effective value of the current is selected to be 4 A. Under the influence of magnetic field superposition of different phases, the composite magnetic field strength is low, and the fluxgate sensor is selected to form an array. Figure 9 shows a picture of the magnetic field strength distribution experimental setup.

The phase current source connected to the three-phase four-wire power cable, cable and three-phase resistor star connection, auxiliary centering device and cable interference fit, and the sensor array fixed, regulated power supply module for the sensor to provide DC 5 V voltage, voltage probe, and Tektronix oscilloscope acquisition sensor two signal. Table 4 shows the experimental instrument and performance parameters.

A comparison of the experimental and simulation results is shown in Figure 10.

In Figure 10, the red curve shows the simulated value of the magnetic field strength in the tangential direction around the cable, and the blue dotted line shows the 50 Hz frequency component of the sensor signal extracted by FFT transformation. As can be seen from Figure 10, the realistic magnetic field distribution is in high agreement with the simulation results, verifying that the locations of the amplitude minima are a = 0°, a = 90°, and a = 270° in the monotonic interval, indicating that the online calibration method proposed in Section 2 is reliable.

### 4.2. Experiment of Current Reconstruction

In order to reduce the experimental chance, and to verify that the simulation conclusions are not affected by the amplitude of the current to be measured, the two experiments used different sizes of current. The current size used in this experiment was 5 A. Using the online calibration method validated in the previous experiment, the sensor array was positioned, as shown in Figure 5, to collect the signal and bring the signal into Equation (1) to decouple and restore the three-phase current waveform to be measured. Figure 11 shows the three-phase source output current waveform with the reconstructed current waveform obtained after online calibration.

From Figure 11 it can be seen that the online calibration method proposed in this paper can be used to restore the current waveforms of each phase. Further calculations show that the amplitude errors of phase A, phase B, and phase C at 50 Hz are 1.874%, 0.593%, and 2.62%, respectively, and the phase angle errors at 50 Hz are 3.731°, 0.136°, and 1.490°, respectively.

## 5. Conclusions

This paper presents a novel method for the online calibration of sensor arrays in order to make it easier to apply sensor modules and handheld measuring devices to the primary equipment in operation in the case of three-phase four-wire power cables in low-voltage distribution networks.This method measures the external tangential magnetic field strength distribution of the cable by rotating the sensor array without destroying the insulation, and finally determines the geometry parameters in the decoupling matrix after finding the four-phase core position by means of the eigenvalues. It has been demonstrated that this method can assist in solving the magnetic field coupling matrix under cable operating conditions, without the aid of calibration currents, and thus restore the phase current waveforms, and is not affected by disturbances such as cable radius, current amplitude, and high-frequency harmonics. Unlike current calibration methods, the magnetic field decoupling matrix in this paper is obtained through theoretical calculations, and therefore requires a high degree of accuracy in the measurement of geometric parameters such as the cable diameter, the radius of the sensor array, and the coaxial of the cable and sensor array, so improving the accuracy of the geometric parameters is the key to further reducing measurement errors.

## Figures and Tables

**Figure 1 sensors-23-02391-f001:**
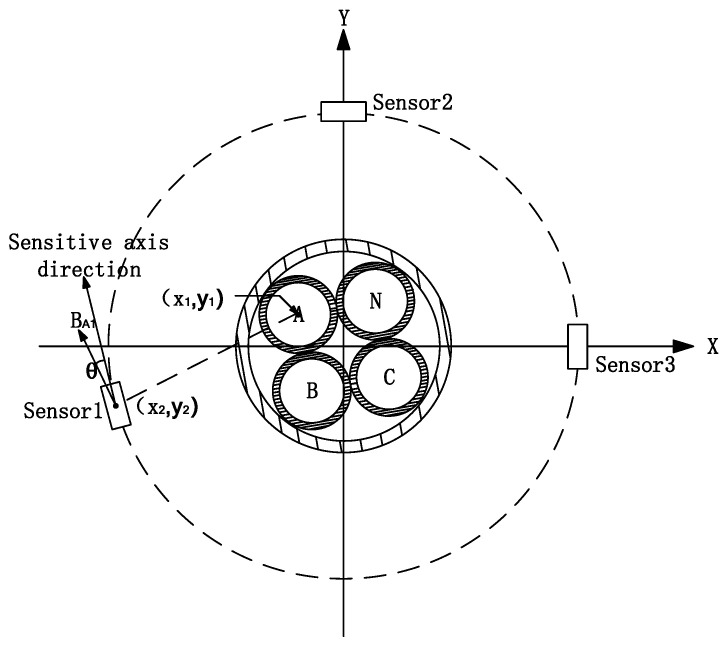
Diagram showing the relationship between the three-phase four-wire power cable and the location of the sensor array.

**Figure 2 sensors-23-02391-f002:**
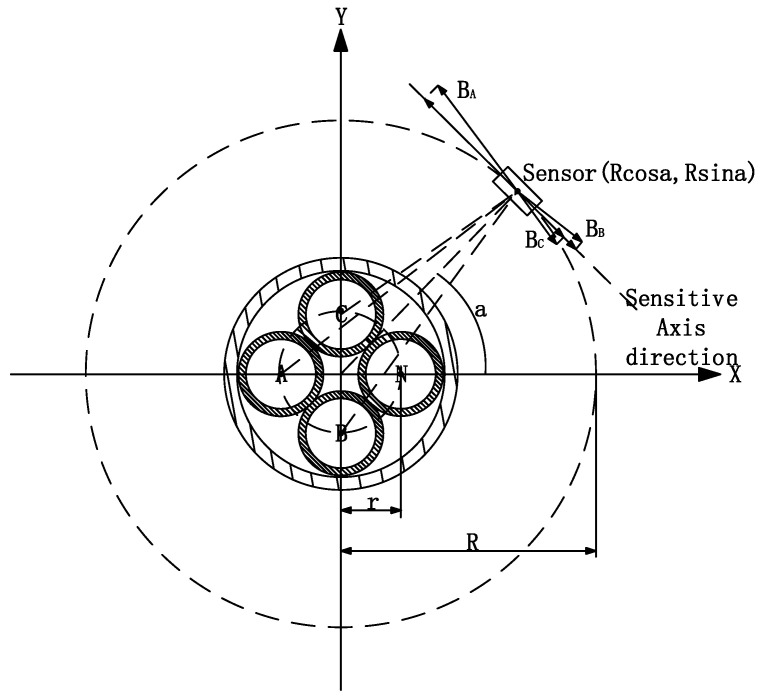
Diagram of the component of the magnetic field strength in the direction of the sensor’s sensitivity axis at a given moment.

**Figure 3 sensors-23-02391-f003:**
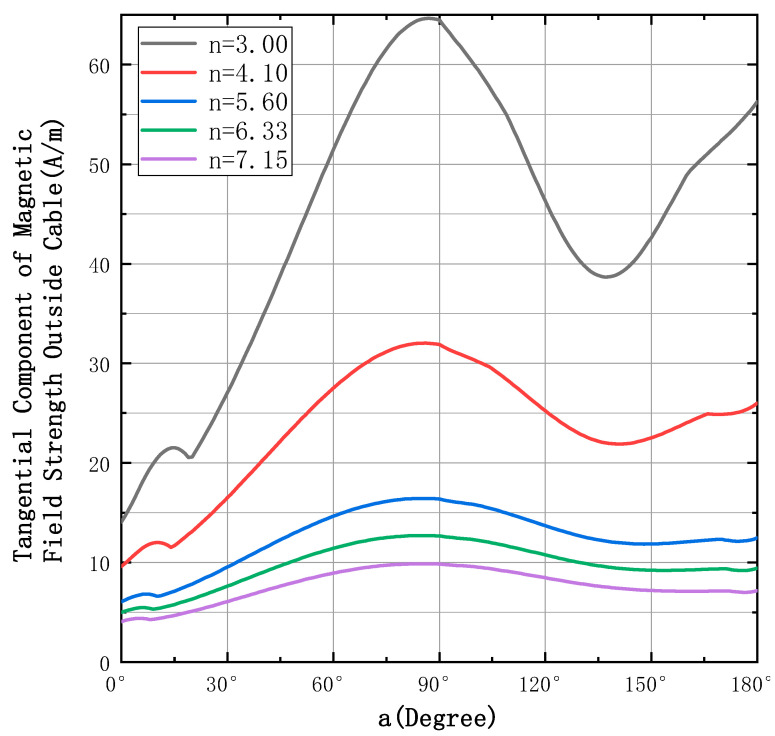
Plot of magnetic field strength amplitude versus “*a*” in the direction of the sensitive axis.

**Figure 4 sensors-23-02391-f004:**
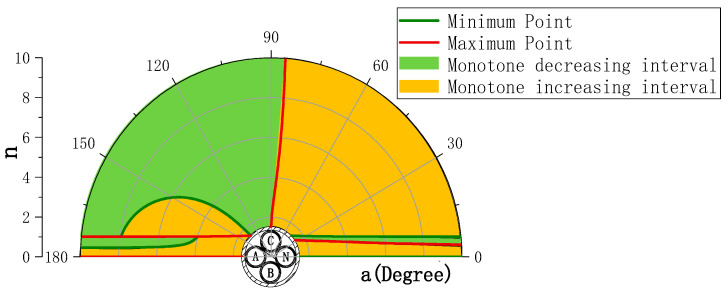
Simulation of the distribution of extremes, and monotonic intervals with “n”.

**Figure 5 sensors-23-02391-f005:**
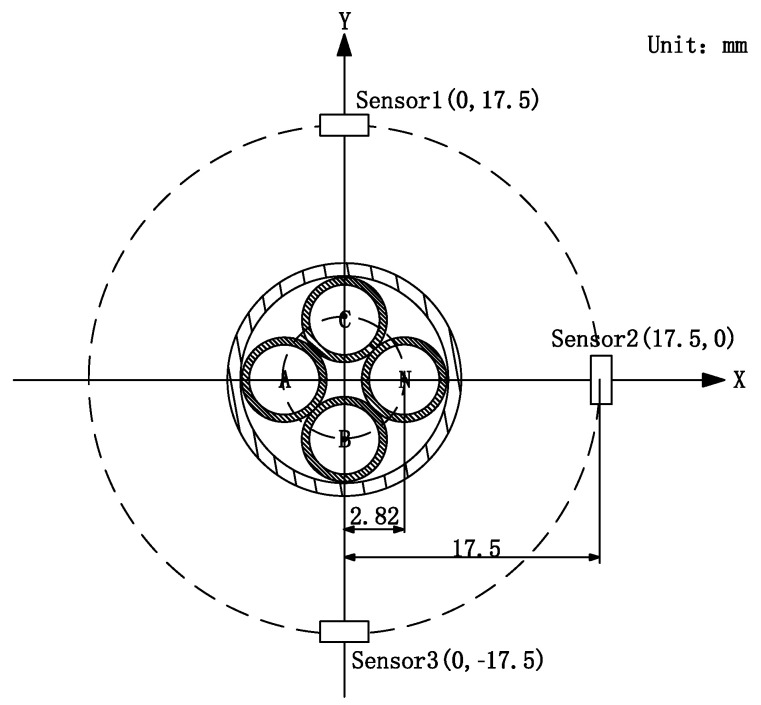
Sensor array design and physical drawing.

**Figure 6 sensors-23-02391-f006:**
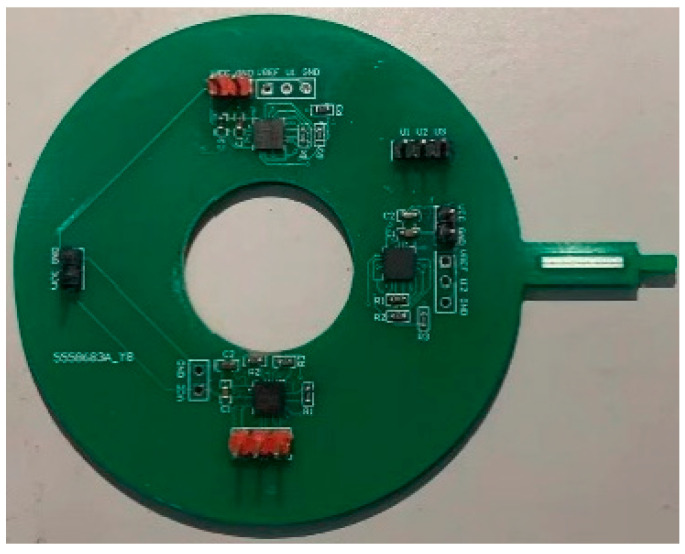
Physical view of the fluxgate sensor array.

**Figure 7 sensors-23-02391-f007:**
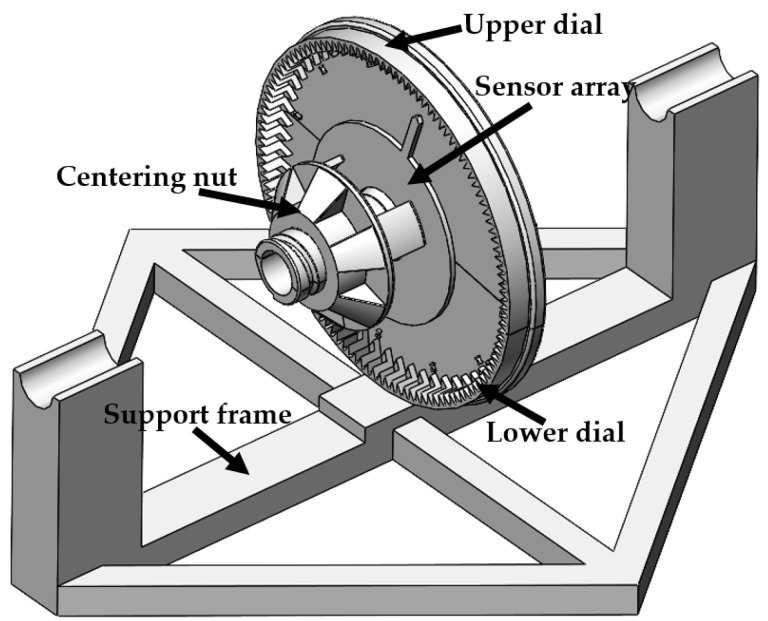
3D model diagram of the auxiliary alignment device.

**Figure 8 sensors-23-02391-f008:**
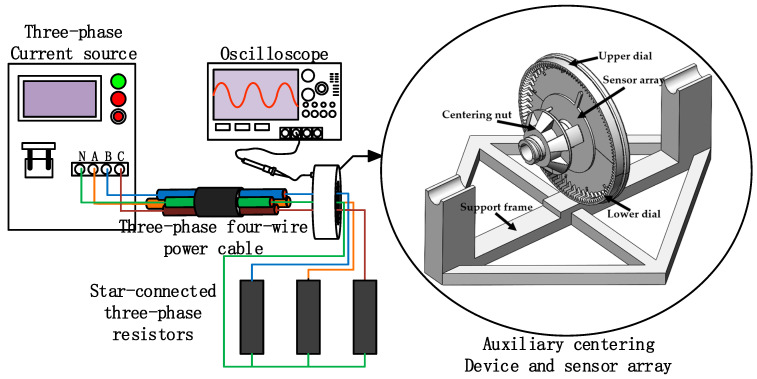
Schematic diagram of the magnetic field strength distribution experimental system.

**Figure 9 sensors-23-02391-f009:**
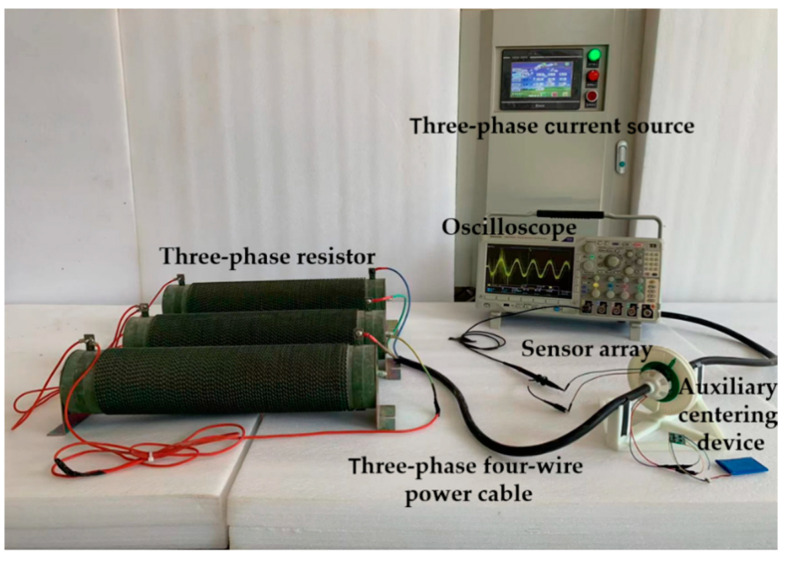
Physical diagram of the magnetic field strength distribution experiment.

**Figure 10 sensors-23-02391-f010:**
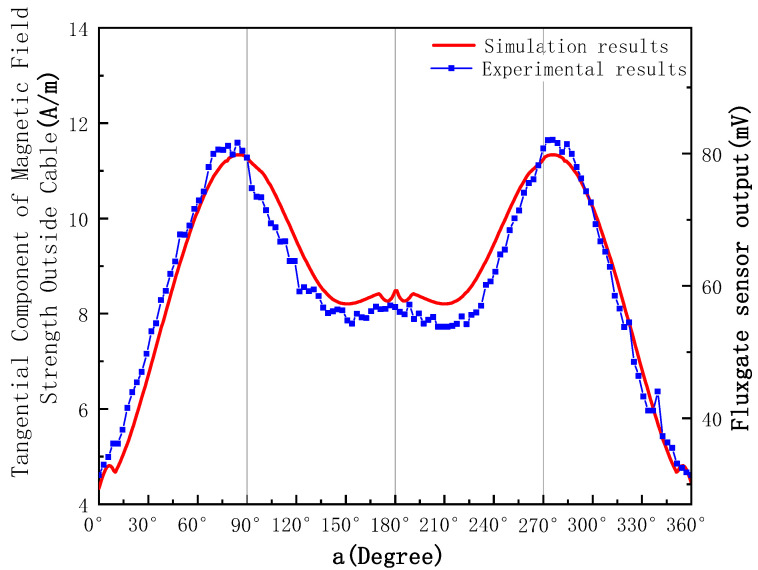
Comparison of experimental and simulation results of magnetic field distribution.

**Figure 11 sensors-23-02391-f011:**
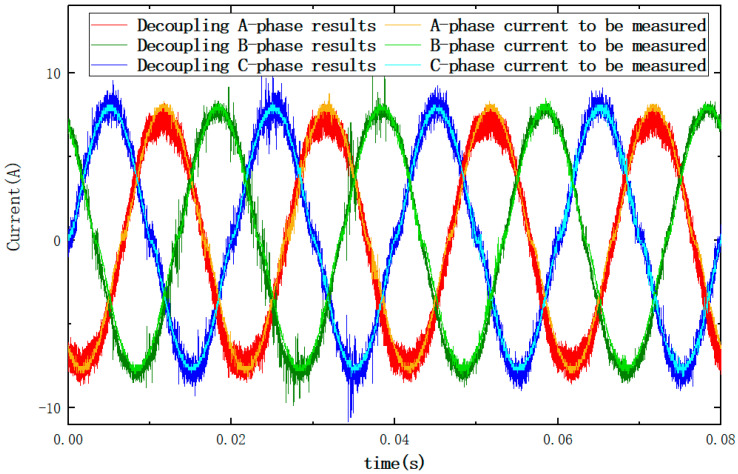
Reconstruction current versus input current graph.

**Table 1 sensors-23-02391-t001:** Distribution pattern of extreme value points.

Extreme Point	*n* = 3	*n* = 4.1	*n* = 5.6	*n* = 6.33	*n* = 7.15
First minimum point	0°	0°	0°	0°	0°
First maximum point	16°	13°	8°	7°	6°
Second minimum point	21°	15°	12°	11°	9°
Second maximum point	89°	88°	86°	85°	86°
Third minimum point	139°	142°	149°	152°	162°
Third maximum point	\	167°	170°	172°	173°
Fourth minimum point	\	172°	177°	179°	179°
Fourth maximum point	180°	180°	180°	180°	180°

**Table 2 sensors-23-02391-t002:** Simulation result.

Simulation of Magnetic Field Strength Distribution Law	
Rule one	the minimum value occurs at a = 0°, the position directly opposite the N phase.
Rule two	the maximum value occurs in the interval of *a* ∈ [−90°, −85°) ∪ (85°, 90°].
Rule three	*a* ∈ [0°, 180°] or *a* ∈ [180°, 360°] there are four extreme value points and four minimal value points in the interval.
Rule four	the positions of phases B and C are in the monotonic interval.

**Table 3 sensors-23-02391-t003:** Distribution law of extreme points.

Minimum Point	
First minimum point	*a* = 0°
First maximum point	$a=\pm \arctan\left(\frac{1}{\sqrt{\left(n^{2}-1\right)}}\right)$
Second minimum point	Between the second and third maximum values
Second maximum point	Between the third and fourth maximum values
Maximum Point	
Third minimum point	Between the first and second minimum values
Third maximum point	*a*∈ [−90°, −85°) ∪ (85°, 90°]
Fourth minimum point	$a=180{}^{\circ}\pm \arctan\left(\frac{1}{\sqrt{\left(n^{2}-1\right)}}\right)$
Fourth maximum point	a = 180°

**Table 4 sensors-23-02391-t004:** Experimental instruments and performance parameters.

Experimental Apparatus	Introduction
Three-phase current source	WuXi Electronic Technology Co., Ltd. ANZ13-3KVA-1000 Hz, maximum output current 6 A
Three-phase four-wire power cable	4 × 6 mm^2^, GB/T5013.4-2008
Oscilloscope	Tektronix MDO3024
Three-phase resistor	10 ± 0.5 Ω
Auxiliary centering device	Photosensitive resin 3D printing, molding size tolerance ±0.2 mm
Fluxgate chip DRV425RTJT	TEXAS INSTRUMENTS

## Data Availability

The study did not report any data.

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
