# Peer review of "Online Calibration Study of Non-Contact Current Sensors for Three-Phase Four-Wire Power Cables"

_sensors, 2023, doi:10.3390/s23052391_

Round 1

Reviewer 1 Report

This manuscript addresses the problem that calibration currents are not easily electrified during transport three-phase four-wire power cables measurements, and proposes a method for obtaining the magnetic field intensity distribution in the tangential direction around the cable, finally enables online self-calibration. Theoretical analysis and simulation calculations show that the method can self-calibrate the sensor array. The accuracy of the simulation is further demonstrated by the experimental results.

The whole manuscript is well stated. The logical and the experimental results agree well with the simulation. It has some practical value, but slightly lacks innovation.

Overall, this manuscript will be appropriate for publication in Sensors, but the following question needs to be modified before acceptance.

1.       Chinese actually appears in Figure 1 and Figure 5. Please fix it carefully.

Therefore, this manuscript is recommended to be accepted with minor revisions.

Reviewer 3 Report

The submitted article for review discusses the subject of Online Calibration of Non-contact Current Sensors for Three-phase Four-wire Power Cables. the topic of the article is supportive. The issue finds use mainly in the field of power electrical engineering. The article is written in a clear and understandable form. The authors process the given problem with a clear goal and a scientifically valuable output. the article has a logical structure and form. the conclusions are supported by the achieved results. Despite the mentioned positives, there are specific shortcomings in the article that need to be eliminated, or revise them. These are the specific comments: 1. It is necessary to unify the term Magnetic Field Intensity and the term Magnetic Field Strength throughout the article. The authors are probably talking about the same quantity, but it is indicated by different word combinations. 2. Fig.3: on the horizontal axis, it is necessary to mark the angle size appropriately, for example with the Alpha symbol, indicating the relevant unit - degree or radian. In the article it is given as the unit "angle", which is not correct. 3. Table 2: in the entire article, it is necessary to unify the designation for "arcus tangent". The authors use the designation "actan" in the table, while the customary designation is "arctan". 4. Fig. 5: there are Chinese characters in the image description. Please correct. 5. Fig. 6: are the used fluxgate sensors single-axis or multi-axis? It would also be appropriate to include the calibration curves of the sensors to see if their response is linear or not. 6. Fig. 8: in the given wiring diagram, the color markings of the individual wires /especially the yellow color/ are poorly visible. If possible, please use a different color scale or use different types of lines. 7. Fig. 10: the fluxgate sensor output is displayed on the horizontal axis: is it an absolute value /because the excitation is harmonic/, or what quantity is it? please add to the text. 8. It is desirable to supplement the list of literature in accordance with the standards of the journal. It is desirable to incorporate the mentioned caveats, because not incorporating them reduces the quality of the submitted article. In conclusion, I conclude that the article requires MAJOR REVISIONS.

Reviewer 4 Report

1. The research motivation of the work is not clear, the more detailed background should be added to the introduction.

2. What is the difference between the detection system proposed in this article and the Hall sensor?

3. The experimental verification provided in the current work is too simple, and a comparative analysis with the reported article should be added.

Round 2

Reviewer 2 Report

The revised paper addressed my concerns regarding the first version.

Reviewer 3 Report

The authors have incorporated the necessary changes into the article. Corrections were carried out at an adequate level. Based on a re-examination of the article, I recommend it for publication.